# Experimental Investigation into the Influence of the Process Parameters of Wire Electric Discharge Machining Using Nimonic-263 Superalloy

**DOI:** 10.3390/ma16155440

**Published:** 2023-08-03

**Authors:** Teshome Dereje, Sivaprakasam Palani, Melaku Desta, Robert Čep

**Affiliations:** 1Department of Mechanical Engineering, College of Engineering, Addis Ababa Science and Technology University, Addis Ababa P.O. Box 16417, Ethiopia; 2Department of Machining, Assembly and Engineering Metrology, Faculty of Mechanical Engineering, VSB-Technical University of Ostrava, 708 00 Ostrava, Czech Republic

**Keywords:** superalloy, WEDM, Taguchi—grey approach, material removal rate, surface roughness, kerf width

## Abstract

Nimonic alloy is difficult to machine using traditional metal cutting techniques because of the high cutting forces required, poor surface integrity, and tool wear. Wire electrical discharge machining (WEDM) is used in a number of sectors to precisely machine complex forms of nickel-based alloy in order to attempt to overcome these challenges and provide high-quality products. The Taguchi-based design of experiments is utilized in this study to conduct the tests and analyses. The gap voltage (GV), pulse-on time (Ton), pulse-off time (Toff), and wire feed (WF), are considered as the variable process factors. GRA is used for the WEDM process optimization for the Nimonic-263 superalloy, which has multiple performance qualities including the material removal rate (MRR), surface roughness (SR), and kerf width (KW). ANOVA analysis was conducted to determine the factors’ importance and influence on the output variables. Multi objective optimization techniques were employed for assessing the machining performances of WEDM using GRA. The ideal input parameter combinations were determined to be a gap voltage (GV) of 40 V, a pulse-on time (Ton) of 8 µs, a pulse-off time (Toff) of 16 µs, and a wire feed (WF) of 4 m/min. A material removal rate of 8.238 mm^3^/min, surface roughness of 2.83 µm, and kerf width of 0.343 mm were obtained. The validation experiments conducted also demonstrated that the predicted and experimental values could accurately forecast the responses.

## 1. Introduction

WEDM is an electro-thermal novel machining technique that uses physical principles to remove material and achieve melting and vaporization by creating regulated, discrete sparks between conductive work surfaces and a wire electrode tool. A dielectric fluid separates and cools the workpiece and removes the eroded particles [1,2,3]. WEDM is commonly used to trim plates ranging in thickness from 1 mm to 300 mm [4]. However, WEDM operation is impacted by various variables, including the workpiece material, the wire electrode, the dielectric medium, and adjustable parameters. Even for a highly competent operator using a cutting-edge WEDM machine, reaching the best performance is rarely achievable due to numerous factors and complicated operation characteristics. The best parameters for a given set of input parameters are determined by modelling the process using appropriate mathematical approaches, which is an efficient way to address this problem [5]. A WEDM computer-controlled positioning system continually regulates the space between the workpiece and wire electrode, which varies from 0.025 to 0.075 mm [2,6]. In the modern and technical world, manufacturing is changing dramatically due to customers’ increased needs for high-quality, dependable, and better components and products. To address these expectations, companies worldwide have begun to focus on lower-cost solutions in machined components and producing items to maintain their profitability [7,8]. Materials with a high hardness, toughness, impact resistance, strength-to-weight ratio, light weight, outstanding resistance against corrosion, and many other attributes are required in today’s mechanical engineering environment.

Throughout the 1950s, creating nickel-based superalloys was a tremendous accomplishment [9]. As a result of their superior qualities and benefits over titanium-based superalloys, industry sectors now regard nickel-based superalloys as the best common service materials [10]. Nevertheless, these alloys are hard to machine because they can maintain their mechanical properties at high temperatures. Due to their insufficient thermal diffusivity, they can create high temperatures near the tool tip and significant variations in temperature within the cutting tool, reducing the tool’s lifespan [9,11,12]. Nimonic superalloy is a nickel-based superalloy that performs well at high temperatures (815 °C) in terms of creep, fatigue, surface oxidation, and corrosion resistance. It also possesses good mechanical characteristics, such as greater hardness and tensile strength.

Because of these characteristics, the aerospace and automotive industries consider Nimonic alloys as the ideal material [12,13]. The characteristics of Nimonic alloys make them a challenge to process. There are several drawbacks to machining Nimonic alloys, including resistance to sustained chip generation during machining operations because Nimonic alloys can retain their properties at high temperatures; high-temperature machining generates sub-par machined surfaces because of the material’s low thermal diffusivity, which causes a high temperature to be created at the tool tip [1,13,14]. It may be possible to process these superalloys using non-convectional manufacturing techniques like wire electrical machining (WEDM). Among those techniques, WEDM is highly suited for fabricating gas turbine blades because it allows for greater freedom in cutting difficult shapes with excellent precision. WEDM also needs less cutting force to remove material, reducing the residual stresses in machined workpieces [15].

Bhupinder Singh and J.P. Misra [16] studied WEDM with a varying peak current, voltage, pulse-off time, and pulse-on time. The best settings for maximizing the MRR and decreasing the SR were determined by utilizing the RSM-based desirability method, and a neural network was then trained to evaluate the impact of variables such as the peak current and the spark gap voltage. Furthermore, the mechanisms of recast layer development and the machined components’ micro-hardness were comprehensively investigated. Significant machining factors were chosen by utilizing decision-making systems with different considerations, which were then enhanced using a special method for sequence selection based on their closeness and optimal responses [17].

Harvinder Singh et al. [18], in order to analyze the surface roughness of Nimonic-75 alloy, attempted a mathematical regression analysis of WEDM. As per their experimental results, the significant factors were Ton, which was followed by Ip. This study examined the RSM and the desirability function techniques for multi-response optimization. Mouralova et al. [19] carried out a detailed design of tests in their study to improve the performance of machining Nimonic-263 by utilizing WEDM. Machining factors such as the pulse-on time, wire feed, gap voltage, discharge current, and pulse-off time were utilized as the input factors. The outcome was that they obtained extensive knowledge regarding the behavior of such machined surfaces, allowing the entire machining process to be optimized. A complete investigation of the prevalence of surface or subsurface concerns was also provided.

B. Singh and Misra [6] studied the machining of Nimonic-263 alloy during the WEDM process to build the best predictive empirical model of the wear rate ratio. Their research found that the process input factors influenced the process performance characteristics significantly. B. Singh and Misra [20] investigated a mathematical model for the cutting speed while improving the WEDM performance of producing nimoic-263 alloy by regarding Ton, Toff, Sv, and Ip as the factor variables. The evaluation of the findings indicated that the essential input variable for the cutting speed was Ton, which was followed by Ip. However, factors such as the pulse-off time (Toff) and servo voltage (SV) had a minimal effect on responses.

B. Singh and Misra [21] developed an empirical model of the cutting speed for WEDM process optimization regarding the factors spark on (Son), peak current (Ip), and servo voltage (Sv) using an RSM-based mathematical model for forecasting the ideal variable settings for the cutting speed. The researchers concluded that if the servo voltage is too small, the removed material and tool wire may not be properly cleaned by the dielectric, resulting in excessive electric arcing and breakage of the wire electrode. Vikasa et al. [22] demonstrated the influence of different input factors on the surface roughness while conducting WEDM to produce EN41 material. The surface roughness parameter was shown to be more significantly impacted by the discharge current. The other parameters’ impacts were noticeably smaller and could be disregarded.

Alias et al. [23] tried to identify the ideal machining factors for WEDM efficiency, such as the material removal rate, kerf width, and surface roughness. In the current work, the machining feed rate was determined to be a critical element. The output of this investigation will improve the efficiency of titanium Ti-6Al-4V products while also cutting machining costs to maximize their economic potential. Bisaria and Shandliya [24] investigated a Nimonic-263 superalloy material during the WEDM process. The experiments employed one-factor-at-a-time methods to estimate the impact of machining factors on the tested component. Their work considered several variables, including the topography, surface morphology, surface roughness, and recast layer thickness.

Mandal et al. [25] developed and presented the key concepts of enhancing the surface integrity of Nimonic-263 superalloy by conducting the WEDM technique and utilizing various post-processing procedures such as etching and grinding. This procedure proved to be an extremely effective and quick way of eliminating all the layers and generating a smooth surface. Taguchi’s parametric design method can be utilized to improve WEDM performance. An experiment was conducted by considering the influence of different cutting factors such as the pulse-on time, gap voltage, pulse-off time, and wire feed on EN31 steel materials. The investigation concluded that the pulse-on time had a bigger impact on the MRR than the other parameters [26].

Divya et al. [27] studied the Taguchi–grey approaches to examine the multi-efficiency improvement of machining variables for the WEDM production of Inconel 800 alloy. Machining factors such as the pulse-on time, peak current, pulse-off time, and gap voltage were considered. The grey relational analysis was employed to discover the best combination of machining variables. These studies concluded that the pulse-on time influenced the machining efficiency characteristic the most. Sheth et al. [28] investigated Inconel 718 alloy by combining the Taguchi–grey relational analysis techniques to achieve the multi-objective optimization of WEDM. The experiment was conducted by utilizing Taguchi’s orthogonal array design by considering the pulse-on time, pulse-off time, and peak current as the input factors on the rate of material removal and the surface roughness. According to the experimental results, the pulse-on time had more of an impact on the material removal rate, whereas the peak current was the essential variable that impacted the surface roughness.

According to the literature, there has not been much research conducted on Nimonic-263 alloy machining. There has not been much research conducted on the application of optimization in Nimonic-263 alloy machining, specifically in terms of the MRR, surface roughness, and kerf width. Hence, exploring the machining performance of WEDM on Nimonic alloys is greatly needed for the current scenario. The novelty of this research lies in investigating the effects of the input process parameters on the properties of Nimonic alloy materials using WEDM process. The influence of the machining factors on the machining performance are needed for modern industry, particularly in terms of achieving a higher material removal rate, a good dimensional accuracy, and a superior surface finish. Therefore, the main objective of the current study is to investigate the machining performance such as the material removal rate, surface roughness, and kerf width of Nimonic alloy using the wire electric discharge machining process.

## 2. Materials and Methods

Gas turbines are typically made of materials that are resistant to high temperatures, such as Nimonic alloys, Inconel alloys, etc. These nickel-based alloys can be precipitation-hardened, allowing them to maintain their strength at high temperatures [29]. Because of its extensive use in marine applications, aerospace applications, the hot-section components of gas turbines, and jet engines and due to its ability to endure very hot operating conditions ranging from 540 °C to 1000 °C without sacrificing its strength, a commercially available [21] Nimonic-263 superalloy was chosen as the workpiece material for the current investigation.

Due to their remarkable strength-to-weight ratio and its resistance to corrosion at high temperatures, nickel-based alloys find several uses in the aerospace, automotive, biomedical, and military industries. Nimonic-263 is reputed for having good process properties as well as promising mechanical properties, a low sensitivity to segregation, a high workability, and a high weldability. It also possesses an excellent elevated temperature strength, creep resistance, and high corrosion and oxidation resistance [30]. The machining of these alloys using conventional process is associated with many problems. Unconventional machining processes like the wire electric discharge machining process are usually used to machine these alloys. The two mechanisms for strengthening that take place in the production of Nimonic-263 are solid solution strengthening and precipitation hardening. Due to the precipitation of the gamma-prime phase during the precipitation hardening heat treatment, precipitation hardening is the key strengthening mechanism in the production of this alloy. Due to their different atomic sizes from the matrix, solution strengthening elements like molybdenum enable high-temperature strengthening through lattice distortion [30].

A commercially available Nimonic-263 superalloy working material plate with a 160 × 10 × 5 mm cross-section was used. The chemical composition of Nimonic-263 superalloy is shown in Table 1. The workpiece was machined with a brass wire electrode with a 0.25 mm diameter, and the dielectric medium was deionized water.

### 2.1. Experimental Setup

The experiments were carried out using a WEDM device (EXCETEK). An Excetek CNC Wire EDM Machine (Model EX 40, Taichung, Taiwan) with a 5-axis closed-loop CNC control device was used for accurate positioning, and the best surface finish that could be obtained using the machine was 0.49 µm Ra. The maximum cutting speed of the machine was 190 mm^2^/min. This configuration had four key sub-elements: (a) a power generator system, (b) a positioning system, (c) a dielectric system, and (d) a drive system [2]. The machine’s effectiveness depended on the essential configuration of the input parameters and the proper setting of the dielectric medium, the diameter, and the kind of wire electrode that was utilized. The efficiency of this machine was determined in the case of the MRR, SR, and KW. The performance of this machine varied depending on the wire speed. A higher cutting speed improved the MRR performance but not the KW and SR. Figure 1 shows the WEDM process setup for machining the Nimonic-263 alloy in this study.

### 2.2. Measurement of Responses

The material removal rate is a recommended WEDM quality that should be set as high as is feasible to maximize productivity and determine the overall cost of the machined surface [31]. A greater MRR number suggests a quicker production rate, which is significantly required in the current sectors. A greater MRR is preferable for the WEDM process [7]. The MRR is determined using Equation (1) [6]:(1)$$\mathrm{MRR}\;({\mathrm{mm}}^{3}/\min)=\frac{Volume\;of\;material\;removed}{Machining\;Time\;(min)}$$

A surface roughness tester, i.e., a 3D optical profilometer, was utilized to evaluate the roughness component of the machined surface. Each specimen’s SR was taken at three different places, and the mean was used as the final value. The kerf width is a measurement of the quality of lost materials during the machining process. A 3D optical profilometer was used to assess the kerf width in the current investigation, and the kerf width was recorded at three different places. The mean measured result was then collected for examination. It was measured in millimeters (mm). The machining time is the time that it takes a wire-cut electrical discharge machine to cut a complete path on a workpiece, and it is usually measured in minutes or seconds. A scanning electron microscope (make: JOEL, Tokyo, Japan) equipped with an energy-dispersive spectroscopy (EDS) device was used to determine the surface topography and to conduct an element analysis. The surface location for the observation of the WEDM surface samples is depicted in Figure 2c.

### 2.3. Experimental Design and Factors

The design of the experiment utilized the selection of the parameters that had the greatest influence on the production process and its outputs and the determination of their optimal values [32]. The current study used the Taguchi orthogonal array (OA) approach using the MINITAB 18 Software to arrange the factors impacting the process and their levels for the machining of the Nimonic-263 alloy utilizing the WEDM process. For the current work, based on the literature review, four machining input control factors were considered as the controlling factors, i.e., Ton, Toff, GV, and WF. They were chosen to determine their effects on the MRR, KW, and SR of the Nimonic-263 alloy obtained during the machining. In this study, based on the machining variable factors and their levels, shown in Table 2 an L16 orthogonal array was identified.

Taguchi’s approach can be used to improve a single performance measure [33]. The Taguchi approach uses a loss function to estimate the gap between measured and optimal results. The signal to noise (S/N) ratio is then created utilizing this loss function. Based on the performance, three categories of S/N ratios are available: the higher the better (HB), the lower the better (LB), and nominally the best (NB). For the HB and LB objectives, the loss function (L) is defined as follows:

With an LB feature, the S/N ratio can be stated as per the formula below [34]:(2)$$\mathrm{For}\;\mathrm{the}\;\mathrm{SR}\;\mathrm{and}\;\mathrm{KW},\;\eta \mathrm{ij}=-10\log\;\frac{1}{n}\sum_{k=1}^{n}y_{ij}^{2}$$

With an HB feature, the S/N ratio can be stated as follows [34]:(3)$$\mathrm{For}\;\mathrm{the}\;\mathrm{MRR},\;\eta \mathrm{ij}=-10\log\;\frac{1}{n}\sum_{k=1}^{n}\frac{1}{y_{ij}^{2}}$$
where i is the experiment number (i = 1, 2 … m), j is the output number (j = 1, 2 … p), k is the replicated number (k = 1, 2 … n), y_ij_ is the loss quality value for the *j*th response at the *i*th experiment, and y_ijk_ is the *k*th replicate value at the *i*th experiment of the *j*th output.

## 3. Results and Discussion

This section discusses the machining of the Nimonic-263 alloy under various machining settings and the results obtained. These investigations considered four levels of the gap voltage, wire feed, pulse-on time, and pulse-off time as the input variables. According to the Taguchi orthogonal array design, 16 experimental combinations were randomly selected. The experimental outcomes of the WEDM process used to machine the Nimoic-263 alloy are displayed in Table 3. The impact of the wire feed, gap voltage, pulse-on time, and pulse-off time on the MRR, SR, and KW during the WEDM process of machining the Nimonic-263 alloy are discussed.

### 3.1. Parametric Influence on Material Removal Rate (MRR)

The investigated data for the material removal rate are shown in Table 3. In this case, the Taguchi response table was utilized to estimate the impact of each input variable value on the machining factors. The results for the process factors are indicated in Table 4.

Figure 3 illustrates the outcomes of Taguchi’s analysis methods for the material removal rate. The response graph shows that the MRR of the WEDM process of machining the Nimonic-263 superalloy was lower when the gap voltage (GV) was raised. The spark gap on the workpiece plays an essential role in determining the value of the MRR. In this case, when the gap voltage value was small, the spark distance between the wire and the workpiece was reduced, which caused the spark strength to increase, which rapidly melted the material. When the gap voltage was increased, the spark intensity widened, which caused the spark’s strength to melt less material [14]. When the pulse-on time (Ton) value was raised, the discharging energy supplied to the machining performance also increased, resulting in a powerful explosion that could potentially increase the MRR [4]. Also, longer conduction times allowed more heat to be transferred to the workpiece as the pulse-on time (Ton) was raised. Because of the greater temperature developed, the workpiece was heated more, and the whole surface of the workpiece became harder due to quenching during the subsequent spark-off. This may be why a significant quantity of heat energy created between the wire material and the test sample due to the greater amount of sparking contributed to rapid erosion and hence an increased material extraction rate [7].

Furthermore, Figure 3 depicts the effect of increasing Toff on the MRR in the opposite direction. As Toff increased, the MRR was achieved at a lower level because of the decrease in the amount of sparking due to the lower discharge energy [15]. A break in the spark was provided during the pulse-off period, allowing time for the cleanup of any spark-related debris. The workpiece was also quenched throughout this time, and the reduced pulse-off time resulted in less time for quenching the workpiece, which caused a loss in hardness. On the other hand, when the pulse-off time was brief, the subsequent spark in front of the work component was able to completely cool and quench. When Ton was increased, the pulse duration in a single duty cycle was lengthened, increasing the discharge energy. Conversely, when Toff was raised, the pulse’s remaining duration in a single duty cycle increased, decreasing the discharge energy [2].

As can be seen in Figure 3, the MRR was raised when the wire feed was raised to 2 m/min, which caused the molten material to clean up across the machining zone. Afterwards, the MRR decreased until the wire feed value reached 3 m/min. At the same time, the highest MRR achieved at this wire feed value reached 4 m/min. In this case, wire vibrations were the primary cause of this fluctuation in the MRR value. Wire vibrations tend to become more noticeable when a greater discharge energy is used. In general, wire vibrations occur when several forces are at work during machining operations, such as reaction forces, hydrodynamic forces, electromagnetic forces, and electrostatic forces [35]. The highest MRR was achieved when the gap voltage (GV) and pulse-off time (Toff) were at the minimal level and the pulse-on time (Ton) and wire feed (WF) were at the maximum level, i.e., GV1, Ton4, Toff1, and WF4.

#### ANOVA for Material Removal Rate (MRR)

ANOVA approaches were used to determine the importance of the variables on the material removal rate, kerf width, and surface roughness. ANOVA was utilized to estimate the percentage of each control factor compared to the identified output factors and to determine the R-square values that indicated adequate results [36]. R-square values and adjustable R-square values were used to evaluate the fitness of the data in the investigation.

The influence of the machining factors was evaluated using the F-value and *p*-value. For the ANOVA analyses, a 95% confidence level was chosen. For a crucial influence on the selected response, the *p*-value should not be more than 0.05% at the 95% confidence level [36]. As observed in the ANOVA results shown in Table 5, Ton had the lowest *p*-value and the greatest F-value, indicating that it was the most important variable influencing the MRR, contributing 52.5%. The GV was the next essential influencing input factor, whereas Toff was the least influential in terms of the MRR, as shown in Table 5. A minimal contribution error of 2.1% was achieved, showing that the current data could be utilized to forecast results in the future with a minimal degree of errors.

### 3.2. Parametric Influence on the Surface Roughness (SR)

Surface roughness was analyzed by utilizing the Taguchi approach, and the outcomes are demonstrated in Table 3. The major effect plot for the means of the data demonstrates the influence of the specific factors at different levels of SR, where a smaller-is-better S/N was utilized for the SR measurements since a lower value of SR implies a higher value of surface quality [33]. As a result, a “smaller is better” ratio was utilized to quantify the surface roughness. Because of the thermal energy produced during the WEDM process, the rate of the spark energy and machining time (Ton) directly affected the removal of material.

Figure 4 depicts a schematic illustration of the surface roughness response graph of the Nimonic-263 alloy during the WEDM process. The results illustrated that the SR decreased as the gap voltage increased, which was because raising the voltage resulted in a widened spark gap. As a consequence, the intensity of the spark was reduced while flushing was increased, resulting in micro-craters on the fabricated parts and an improved surface finish [37,38]. On other hand, Figure 4 demonstrates a decrease in the surface roughness as Toff increased. Increasing Toff reduced the number of active sparks, which reduced the discharge energy. An improved surface roughness of the machined component was achieved when less thermal energy was consumed [36]. This was because the high Toff value increased the cooling time, which increased the flushing time, which caused a proportionally greater amount of molten material to flow out over the conducted surface and a decrease in the surface roughness. According to Figure 4, it was seen that surface roughness increased for a short period of time and decreased with the decrease in the wire feed. The surface roughness dropped up to a certain point and then increased in step with the wire feed supply. As can be seen from the graph, the surface roughness was deflected upward and downward due to the occurrence of wire vibrations. Unfavorable sparking conditions were created as a result, which increased the SR by creating irregular craters on the conducted surface [15]. The response table for surface roughness is shown in Table 6.

#### ANOVA for Surface Roughness (SR)

Table 7 shows a statistical study of the SR using ANOVA and demonstrates that some of the machining factors had a substantial influence on the SR with a *p*-value for each variable of less than 0.05. The analysis of variance showed that when compared to the other factors, the pulse-on time and wire feed were essentially the influencing variables affecting the surface roughness. A minor contribution error of 2.165% was found, illustrating that the current data could be utilized to make future estimations with a minimal amount of inaccuracy.

### 3.3. Parametric Influence on the Kerf Width (KW)

The measured experimental values of the kerf width are displayed in Table 3. During the kerf width analysis, the smaller-the-better approach to the performance analysis was utilized. Researchers have not paid enough attention to the kerf width despite the fact that it is a critical response parameter for maintaining the dimensional precision of machined components. During the WEDM process, the kind of electrode wire, the volume of the dielectric fluid used, the material being processed, and, especially, the machine setup factors all significantly affect the kerf width [39]. The Taguchi response (Table 8) was employed in this study to evaluate the effect of each level of the input factors on the machining quality. As shown in Figure 5, the mean value gradually rose as the gap voltage rose. With on and off rising and falling pulses, the mean value was deflected. The mean value fell for a short period and then increased in response to the wire feed. The optimal setting of the kerf width was obtained when the pulse-on time, pulse-off time, and gap voltage were at the initial levels while the wire feed was at the second level, as shown in Table 8.

#### Analysis of Variance (ANOVA) for Kerf Width

A statistical analysis of the KW using ANOVA is illustrated in Table 9. Table 9 shows that the same machining factors that had an essential effect on the KW were determined to be Ton, Toff, WF, and GV, respectively, with some input variables having a fit of less than 0.05 for the *p*-values. According to the results of the ANOVA of the kerf width, the gap voltage was the most influential control factor, and the wire feed was the least influential factor during the WEDM process of machining the Nimonic-263 alloy when compared to the other parameters. The present data were found to have a minimal error contribution of 1.72%, indicating that they could be used to produce future estimations with the a minimal level of error.

### 3.4. Surface Integrity Analysis

Using SEM (make: JOEL) and an accelerating voltage of 20.0 keV, Figure 6a–c depict the surfaces produced using the WEDM process at a 100× magnification. For the purpose of observing the microstructures of the surfaces obtained using the WEDM process at a low discharge energy (LDE), medium discharge energy (MDE), and high discharge energy (HDE), three specimens were chosen. It is clear from the study of the specimens that the surfaces obtained using the WEDM process presented with microcracks, craters, and debris made of molten metal.

Based on the findings, it was concluded that the surface topography depended on the volume of energy released during the discharging process. The surface state obtained using the WEDM process using LDE when the parameters were adjusted to their minimum values is depicted in Figure 6a. The surface appeared smoother with this parametric setting because there were fewer surface micro-voids caused by using LDE and extremely small discharge craters with less debris. The surface roughness, Ra, obtained using these machining conditions was 2.607 µm, and the kerf width value was 0.3261 mm. The surface obtained using MDE, where deep and wide craters were obtained on the surface obtained using the WEDM process, is shown in Figure 6b. The surface finish became rougher (Ra = 3.112 µm), the material removal rate (MRR) increased to 5.565 mm^3^/min, and the kerf width increased to 0.3360 mm using MDE. Figure 6c clearly illustrates the surface using HDE. As the discharge energy increased, larger and deeper craters, pockmarks, and debris lumps developed. Figure 6c makes it clear that the size of the bombard was deeper and wider due to the high Ton (8 µs) and 70 V, which increased the pulse discharge energy on the cutting zone and ultimately led to a high SR (3.815 µm).

Figure 7 shows that EDS analysis of the surfaces obtained using the WEDM process revealed that copper and zinc residues from the wire tool material were deposited on the surface. The workpiece surface and the brass wire tool evaporated, melted, and then solidified again as a result. The EDS analysis also demonstrated that components from the dielectric fluid were diffused into the machined surface, causing oxidation on the surfaces obtained using the WEDM. So, by decreasing the amount of alloying elements in the working material, the characteristics of components obtained using the WEDM process could be impacted.

### 3.5. Multi-Parametric Optimization by Using Taguchi—Grey Relation Analysis

Deng (1982) developed the grey relation analysis (GRA) method to meet the essential mathematical requirements for working with a poor, imperfect, and unsure system. The grey relational grade is evaluated by using GRA to assess various performance characteristics. As a result, an improvement in a single grey relational grade replaces the optimization of multiple performance attributes. The following measures are conducted to optimize the MRR, SR, and KW while simultaneously utilizing grey relational analysis (GRA) [40]. The best process parameter settings for a single response characteristic can be found using Taguchi’s experimental method. Multi-response optimization using GRA is the preferred approach when there are two or more responses with different quality attributes. The relationship between potentially erratic finite data can also be determined via grey relational analysis [41]. Therefore, the multi-response optimization of the WEDM parameters was carried out in this work by employing the following GRA stages: the Taguchi-grey relational analysis method and the desirability approach were used for finding the ideal set of parameters in order to increase the material removal rate (MRR) and decrease the surface roughness (Ra) [42,43].

#### 3.5.1. Normalization

Because the experimental data were made up of multiple units/dimensions, a comparison analysis was impossible. As shown in Table 10, grey relational development was used to normalize the investigation output results into similar units, with the “higher the better” and “smaller the better” criteria being used for maximizing and decreasing the output parameters. To create an array between “0” and “1”, an appropriate value was produced from the original value. If the output was to be downsized, the smaller-the-better quality of normalization was designed to reduce the output to tolerable levels. Using the equations of the loss function to obtain the S/N ratio values for the output responses of the MRR, SR, and KW, with greater values for the MRR and smaller values for the SR and KW, this could then be normalized as follows.

In terms of the MRR (the higher-the-better kind of problem) [44],
(4)$$X_{i}^{*}\left(k\right)=\frac{X_{i}\left(k\right)-MinX_{i(k)}}{Maxx_{i}\left(k\right)-Min{xX}_{i}(k)}$$
where i = 1 to m … (m is the total number of datapoints), k = 1 to n … (n is the total number of responses), Xi = the original data before the processing sequence, Xi* is the sequence after the processing is complete, Max. Xi (k) indicates the maximum value of Xi (k), and Min. Xi is the required response.

For the SR and KW (the lower-the-better kind of problem) [44],
(5)$$X_{i}^{*}\left(k\right)=\frac{\mathrm{Max}X_{i}\left(k\right)-X_{i}(k)}{MaxX_{i}\left(k\right)-MinX_{i}(k)}$$

#### 3.5.2. Sequences Deviation, Δ0i (k)

The sequence deviation, Δ0i (k), is the absolute difference between the sequence reference, x0*(k), and the sequential comparison, xi*(k), following normalization [45]. It is calculated utilizing the following formula [45]:(6)$$\Delta_{0i}\;(k)=|X_{0}^{*}\left(k)-X_{i}^{*}(k\right)|$$
where Δ_0i_ = the sequence deviation of the sequence reference, $X_{0}^{*}$(k) = denotes the sequential comparison, $X_{0}^{*}$(k) = 1 is the highest normalizing value, and $X_{i}^{*}\left(k\right)\;$ = denotes the sequential comparison.

#### 3.5.3. Grey Relational Analysis (GRA)

For all sequences, GRC reflects the link between the optimal and the actual normalized S/N ratios. They have one grey relational coefficient if the two sequences match at every location. The GRC is ζi(k) for the *k*th quality of the *i*th characteristic. The determination formula is written as [40]:(7)$$\zeta_{i}(k)=\frac{{∆}_{min}+\zeta {∆}_{max}}{\;{∆}_{0i}\left(k\right)+\zeta {∆}_{max}}$$
where ${∆}_{min}$ = min||$X_{0}^{*}\left(k\right)-X_{i}^{*}(k)$|| = smaller values of Δ_0i_ and ∆_max_ = max||$X_{0}^{*}\left(k\right)-X_{i}^{*}(k)$|| = maximum value of Δ_0i_, ζ = identification coefficient, where ζ ϵ[0, 1] (the value can be ordered on the basis of the effective system requirements). The range of ζ is between the minimum and maximum sense of distinguished ability. ζ = 0.5 is commonly utilized. GRC was used in this study for 16 sequential comparisons.

#### 3.5.4. Grey Relational Grade (GRG)

After determining the grey relational coefficients, GRG typically uses the mean value of the grey relational coefficient as the grey relation grade [45]. GRG is evaluated using the following formula:(8)$$\gamma i=\frac{1}{m}\sum_{k=1}^{n}w\;x\;\zeta i(k)$$
where γi = grey relational grade, m = number of responses, n = number of runs, and w = weight factor. This strategy simplifies the multi-response factor optimization issue into a single-output optimization scenario with an overall GRG as the achievement goal [45]. The experimental results are more closely related to the ideal normalized value as the GRG increases with how closely the related factor combination approaches the ideal. Utilizing the GRG allows for the assessment of the influence of a factor and the determination of the optimal level for each controllable component.

### 3.6. Determine the Optimal Factor and Its Combination of Levels

A single GRG can be used to optimize various performance criteria [46]. The experimental setting of level 4 had the greatest GRG, as shown by Table 10, which shows the GRG values for the process variables at different levels. As a consequence, out of the sixteen total trials, the fourth GRA test experiment had the greatest accomplishments in a variety of areas. Table 11 displays the average value for each GRG parameter range as well as the overall mean of the GRG (0.56041) calculated across all sixteen experiments. Figure 8 demonstrates the Taguchi approach that was utilized to determine the influence of each variable factor on the GRG at various levels. The value of delta is the gap between the highest and lowest GRG values for each specific variable factor and its level, indicating that the larger the delta value, the greater the impact of the variable on the GRG. In general, the greater the value of the GRG, the better the performance of the various features [46]. The set of Gv1, Ton4, Toff4, and WF4 had the highest values for the GRG. As a result, this combination was the best combination of parameters for multi-machining features.

### 3.7. Variance Analysis (ANOVA) for Grey Relational Grade

The ideal values of the machining variables were estimated using the same approach as mentioned in the preceding section. The ANOVA findings shown in Table 12 demonstrated that the pulse-on time (Ton) was the essential parameter influencing the grade values under the 95% confidence level (*p* ≤ 0.05), whereas the pulse-off time (Toff) impacted them less under the 95% confidence level. As a result, only the most important process factors were evaluated to estimate the best values of the machining attributes.

### 3.8. Confirmation Experiment

To examine the quality attributes of the Nimonic-263 alloys machined using the WEDM process, a validation experiment for the ideal variable factors with their chosen range was carried out. The fourth experiment run (Table 10) displayed the largest grey relational grade value, suggesting that the ideal variable set of GV1, Ton4, Toff4, and WF4 had the best performance features out of the sixteen trials. The results of the confirmation experiment were compared with those of the original experiment to confirm the findings. The prediction performance of the created model was good if the percentage of the expected error was less than 5%, as shown in Table 13.

## 4. Conclusions

The machining of Nimonic-263 superalloy using the WEDM process was carried out utilizing the integrated Taguchi–grey relational analysis method to enhance the multi-criteria features of the material removal rate, surface roughness, and kerf width. The following findings were drawn in terms of the WEDM machining process:This study gives insightful information on the optimization of the WEDM process by using GRA for the Nimonic-263 alloy;The ANOVA investigation showed that the pulse-on time (Ton) and gap voltage (GV) had the greatest influence on the MRR at 52.5% and 42.5%, respectively, whereas Toff and the wire feed had less of an effect on the MRR;The pulse-on time, wire feed, and pulse-off time had significant effects on the SR (41.05%, 33.01%, and 18.11%, respectively), whereas the gap voltage had a lesser impact (5.69% of the total contribution) on the SR;The gap voltage and pulse-off time had more of an influence on the kerf width, with percentage contributions of 83.03% and 11.63%, respectively. The pulse-on time and wire feed had a less significant impact on the KW;The optimal set of process variables for the multi-objective optimization using GRA were GV (40 V), Ton (8 µs), Toff (16 µs), and WF (4 m/min);The predicted values produced by the GRA model were found to be quite comparable to the observed values;The percentage error between the actual and predicted results for the MRR, KW, and SR utilizing GRA was 2.975%, 3.51%, and 3.562%, respectively;As a result, by utilizing GRA, the process parameters for different machining characteristics during the machining of Nimonic-263 alloy using the WEDM process can be successfully optimized.

## Figures and Tables

**Figure 1 materials-16-05440-f001:**
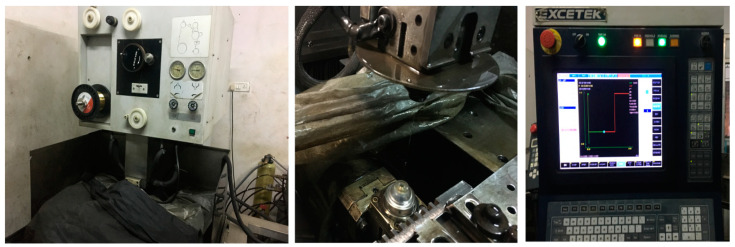
WEDM experimental setup.

**Figure 2 materials-16-05440-f002:**
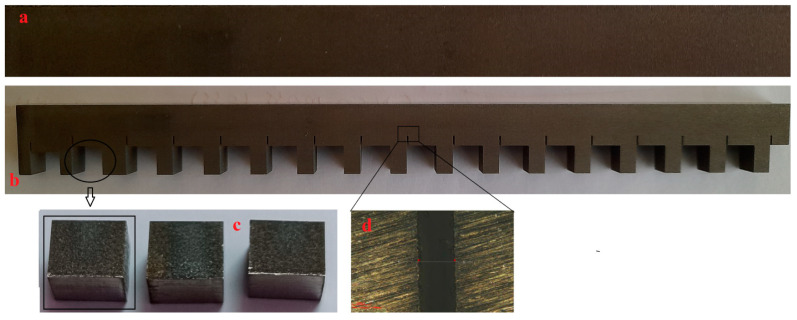
Images of samples (**a**) before machining and (**b**) after machining. (**c**) Samples for SEM and (**d**) kerf width measurement.

**Figure 3 materials-16-05440-f003:**
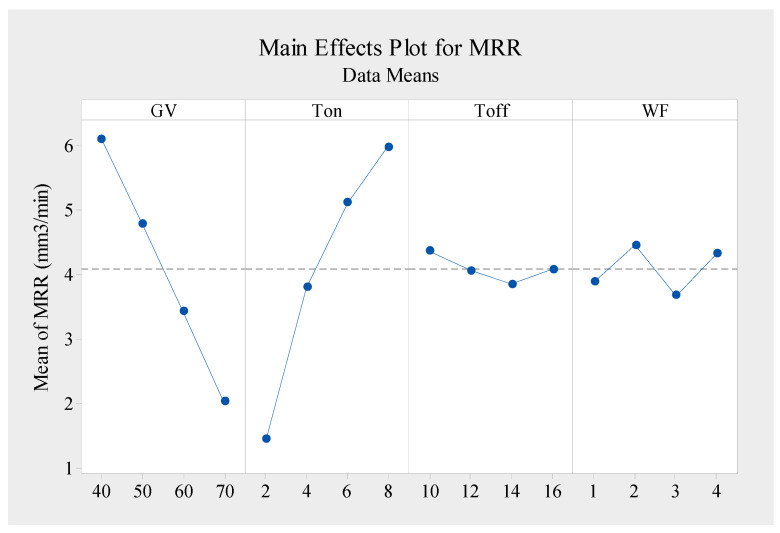
Response graph for data of mean material removal rate (higher is better).

**Figure 4 materials-16-05440-f004:**
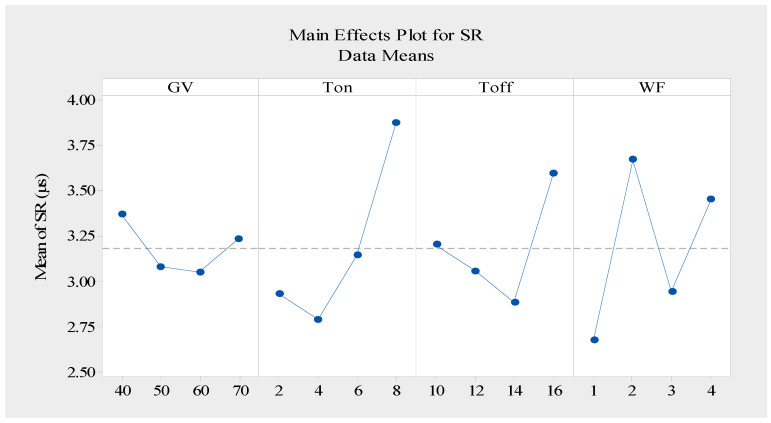
Response graph for means of data of surface roughness (smaller is better).

**Figure 5 materials-16-05440-f005:**
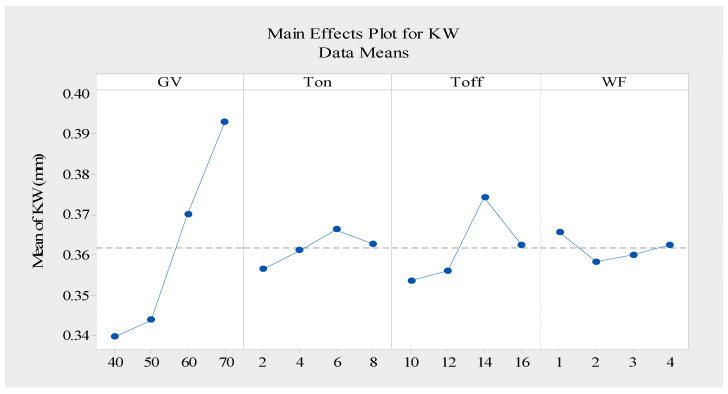
Response graph for means of kerf width data (smaller is better).

**Figure 6 materials-16-05440-f006:**
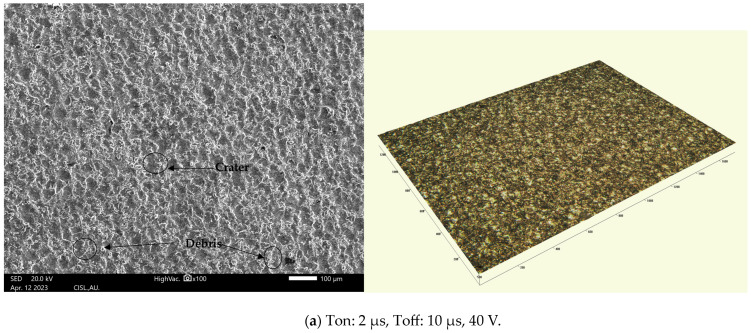
(**a**–**c**) SEM and 3D images of surfaces obtained using the WEDM process.

**Figure 7 materials-16-05440-f007:**
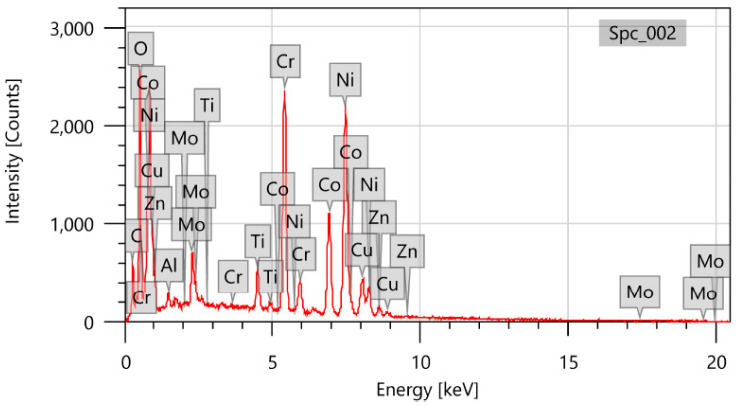
EDS images of machined sample.

**Figure 8 materials-16-05440-f008:**
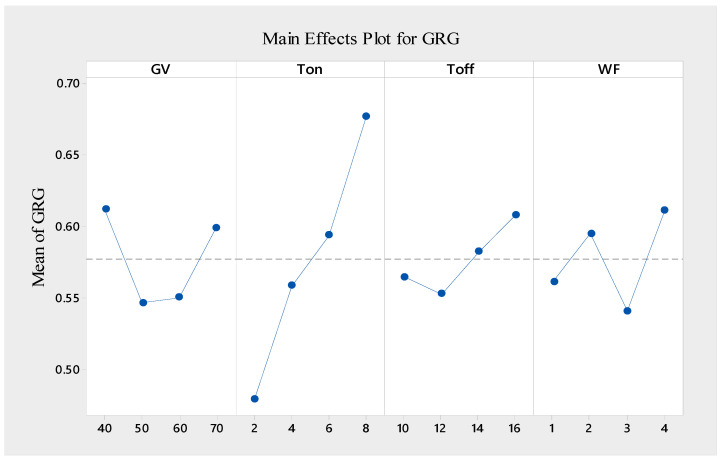
Response graph for mean values of grey relational grade.

**Table 1 materials-16-05440-t001:** Nimonic-263 superalloy’s chemical composition.

Elements	C	Al	Ti	Mo	Co	Cr	Fe	Ni
Weight %	0.045	0.5	2.1	5.6	20	21	0.5	50.27
**Elements**	**Si**	**O**	**S**	**N**	**P**	**Mn**	**Cu**	
Weight %	0.04	0.0022	0.005	0.0031	0.005	0.43	0.002	

**Table 2 materials-16-05440-t002:** Shows variables factors and their levels.

No	Variable	Unit	Range	Levels			
				L1	L2	L3	L4
1	Gap voltage (GV)	Volts	40–70	40	50	60	70
2	Pulse-on time (Ton)	(µs)	2–8	2	4	6	8
3	Pulse-off time (Toff)	(µs)	10–16	10	12	14	16
4	Wire feed (WF)	m/min	1–4	1	2	3	4

**Table 3 materials-16-05440-t003:** Experimental design and values for output responses and S/N ratio results.

	Input Parameters				Response Parameters			S/N Ratio		
Run No	GV (V)	Ton (µs)	Toff (µs)	WF (m/min)	MRR (mm^3^/min)	SR (µm)	KW (mm)	MRR	SR	KW
1	40	2	10	1	3.126	2.607	0.3261	9.899	−8.3228	9.7330
2	40	4	12	2	6.141	3.451	0.3305	15.764	−10.7589	9.6166
3	40	6	14	3	7.040	2.934	0.3554	16.951	−9.3492	8.9857
4	40	8	16	4	8.123	4.496	0.3469	19.007	−13.9289	9.1959
5	50	2	12	3	1.613	2.215	0.3360	4.152	−6.9075	9.4732
6	50	4	10	4	5.565	3.112	0.3360	14.909	−9.8608	9.4732
7	50	6	16	1	5.571	3.056	0.3536	14.918	−9.7031	9.0298
8	50	8	14	2	6.383	3.940	0.3500	16.100	−11.9099	9.1186
9	60	2	14	4	0.773	2.883	0.3786	−2.236	−9.1969	8.4364
10	60	4	16	3	2.305	2.813	0.3645	7.253	−8.9834	8.7660
11	60	6	10	2	5.011	3.272	0.3677	13.998	−10.2963	8.6901
12	60	8	12	1	5.666	3.245	0.3695	15.065	−10.2243	8.6477
13	70	2	16	2	0.328	4.023	0.3852	−9.682	−12.0910	8.2863
14	70	4	14	1	1.210	1.790	0.4136	1.655	−5.0571	7.6684
15	70	6	12	4	2.835	3.314	0.3886	9.051	−10.4071	8.2099
16	70	8	10	3	3.750	3.815	0.3845	11.480	−11.6299	8.3021

**Table 4 materials-16-05440-t004:** The response table for mean material removal rate.

Level	GV	Ton	Toff	WF
1	6.307	1.460	4.363	3.893
2	4.783	3.805	4.064	4.466
3	3.439	5.114	3.851	3.677
4	2.031	6.180	4.281	4.523
Delta	4.276	4.720	0.512	0.846
Rank	2	1	4	3

Delta = highest mean − lowest mean (for each parameter).

**Table 5 materials-16-05440-t005:** ANOVA results for material removal rate.

Source	DoE	Adj.SS	Adj.MS	F-Value	*p*-Value	% Contribution
GV	3	40.1958	13.3986	20.22	0.017 *	42.5%
Ton	3	49.6168	16.5389	24.96	0.013 *	52.5%
Toff	3	0.6346	0.2115	0.32	0.813	0.67%
WF	3	2.1130	0.7043	1.06	0.481	2.23%
Error	3	1.9882	0.6627			2.1%
Total	15	94.5484				
R-sq = 97.90%, R-sq(adj) = 89.49%						

* indicates siginificant term.

**Table 6 materials-16-05440-t006:** The response table with mean values of surface roughness.

Level	GV	Ton	Toff	WF
1	3.491	2.932	3.201	2.675
2	3.081	2.792	3.056	3.672
3	3.053	3.144	2.887	2.944
4	3.235	3.993	3.716	3.570
Delta	0.438	1.201	0.829	0.997
Rank	4	1	3	2

**Table 7 materials-16-05440-t007:** ANOVA results for surface roughness.

Source	DF	Adj.SS	Adj.MS	F-Value	*p*-Value	% Contribution
GV	3	0.4826	0.16086	2.63	0.224	5.692%
Ton	3	3.4775	1.15917	18.94	0.019 *	41.015%
Toff	3	1.5355	0.51184	8.36	0.057	18.11%
WF	3	2.7995	0.93315	15.25	0.025 *	33.018%
Error	3	0.1836	0.06119			2.165%
Total	15	8.4786				
		R-sq = 97.83%		R-sq(adj) = 89.17%		

* indicates siginificant term.

**Table 8 materials-16-05440-t008:** The mean response table for kerf width.

Level	GV	Ton	Toff	WF
1	0.3397	0.3565	0.3536	0.3657
2	0.3439	0.3612	0.3562	0.3584
3	0.3701	0.3663	0.3744	0.3601
4	0.3930	0.3627	0.3625	0.3625
Delta	0.0533	0.0099	0.0208	0.0073
Rank	1	3	2	4

**Table 9 materials-16-05440-t009:** ANOVA results for kerf width.

Source	DF	Adj.SS	Adj.MS	F-Values	*p*-Values	% Contributions
GV	3	0.007392	0.002464	48.23	0.005 *	83.03%
Ton	3	0.000200	0.000067	1.31	0.416	2.25%
Toff	3	0.001035	0.000345	6.75	0.076	11.63%
WF	3	0.000122	0.000041	0.79	0.573	1.37%
Error	3	0.000153	0.000051			1.72%
Total	15	0.008903				
R-sq = 98.28%, R-sq (adj) = 91.39%						

* indicates siginificant term.

**Table 10 materials-16-05440-t010:** Calculated values for normalization, deviation sequence, GRC, GRG, and rank.

Run	Normalized S/N Ratio			Sequence Deviation			GRC			GRG	Rank
	MRR	SR	KW	MRR	SR	KW	MRR	SR	KW		
1	0.6826	0.3681	0.0000	0.3174	0.6319	1.0000	0.6117	0.4417	0.3333	0.4622	15
2	0.8870	0.6427	0.0564	0.1130	0.3573	0.9436	0.8156	0.5832	0.3464	0.5817	9
3	0.9283	0.4838	0.3620	0.0717	0.5162	0.6380	0.8746	0.4920	0.4394	0.6020	5
4	1.0000	1.0000	0.2601	0.0000	0.0000	0.7399	1.0000	1.0000	0.4033	0.8011	1
5	0.4822	0.2086	0.1258	0.5178	0.7914	0.8742	0.4913	0.3872	0.3639	0.4141	16
6	0.8572	0.5415	0.1258	0.1428	0.4585	0.8742	0.7778	0.5216	0.3639	0.5544	12
7	0.8575	0.5237	0.3406	0.1425	0.4763	0.6594	0.7782	0.5121	0.4313	0.5739	10
8	0.8987	0.7724	0.2976	0.1013	0.2276	0.7024	0.8315	0.6872	0.4158	0.6448	2
9	0.2595	0.4666	0.6280	0.7405	0.5334	0.3720	0.4031	0.4839	0.5734	0.4868	14
10	0.5903	0.4426	0.4683	0.4097	0.5574	0.5317	0.5496	0.4728	0.4847	0.5024	13
11	0.8254	0.5905	0.5051	0.1746	0.4095	0.4949	0.7412	0.5498	0.5026	0.5978	7
12	0.8626	0.5824	0.5257	0.1374	0.4176	0.4743	0.7844	0.5449	0.5132	0.6142	4
13	0.0000	0.7928	0.7007	1.0000	0.2072	0.2993	0.3333	0.7071	0.6256	0.5553	11
14	0.3952	0.0000	1.0000	0.6048	1.0000	0.0000	0.4526	0.3333	1.0000	0.5953	8
15	0.6530	0.6030	0.7377	0.3470	0.3970	0.2623	0.5903	0.5574	0.6559	0.6012	6
16	0.7377	0.7409	0.6931	0.2623	0.2591	0.3069	0.6559	0.6586	0.6196	0.6447	3

**Table 11 materials-16-05440-t011:** The response table for mean values of grey relational grade.

Level	GV	Ton	Toff	WF
1	0.6118 *	0.4796	0.5648	0.5614
2	0.5468	0.5585	0.5528	0.5949
3	0.5503	0.5937	0.5822	0.5408
4	0.5991	0.6762 *	0.6082 *	0.6109 *
Delta	0.0650	0.1966	0.0554	0.0701
Rank	3	1	4	2
Total mean of GRG = 0.56041				

* indicates siginificant term.

**Table 12 materials-16-05440-t012:** ANOVA results for GRG.

Source	DF	Adj.SS	Adj.MS	F-Value	*p*-Value	% Contribution
GV	3	0.013288	0.004429	2.62	0.225	11.34%
Ton	3	0.079805	0.026602	15.72	0.024	68.097%
Toff	3	0.006937	0.002312	1.37	0.402	5.92%
WF	3	0.012087	0.004029	2.38	0.247	10.313%
Error	3	0.005076	0.001692			4.33%
Total	15	0.117193				
R-sq = 95.67%, R-sq (adj) = 78.34%						

**Table 13 materials-16-05440-t013:** The predicted and confirmed values at a single ideal setting.

Machining Qualities	Optimal Parameter Combinations	Optimal Predicted Values	Experimental Values	Prediction Error (%)
MRR	GV1, Ton4, Toff4, and WF4	8.238	8.125 mm^3^/min	1.39
SR	GV1, Ton4, Toff4, and WF4	2.83	2.94 µm	3.85
KW	GV1, Ton4, Toff4, and WF4	0.343	0.356 mm	3.562

## Data Availability

The data presented in this study are available in the article.

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
