# Peer review of "Experimental Investigation into the Influence of the Process Parameters of Wire Electric Discharge Machining Using Nimonic-263 Superalloy"

_materials, 2023, doi:10.3390/ma16155440_

Round 1
Reviewer 1 Report
1. Equation 1 - please check the typo
2. Please explain the novelty of the current work.
3. Why is Nimonic-263 super alloy selected for this study? Authors can explain the strengthening mechanism in superalloys.
4. Figure 5: please expand the figures and improve the clarity.
5. Please add the images of samples before and after machining and indicate the area where SEM was taken.
6. Please explain the effect of surface topography (from SEM and 3D images) in detail.
7. Explain the taguchi grey relational analysis with scientific rigour. Authors can refer and cite the following papers in order to strengthen the discussion.
https://doi.org/10.1016/j.optlastec.2022.108210, https://doi.org/10.1016/j.optlastec.2023.109306
8. Conclusions can be made crisp and pointwise.
Minor editing of English language required
Author Response
Dear Reviewer
The authors would like to thank the reviewers for their time spent providing constructive feedback on our manuscript. Please find the attachment for response to reviewer.

Reviewer 2 Report
This article claims that wire electric discharge machining process variable optimization for Nimonic-263 super alloy was demonstrated using a Taguchi approach. Unfortunately, this article has some serious issues, see below.
1. I do not see the novelty or significance in the work presented.
2. I do not see the point of Figure 1.
3. Where is the scanning electron microscopy and energy dispersive X-ray spectroscopy work described in the Experimental section?
4. In Table 3, please write numbers with the same number of decimal places.
5. Please put standard deviation values in Figures 2-4 and 7.
6. Please provide references for the equations used in the text.
There are many grammatical errors in the text that must be fixed.
Author Response

(The authors gave the same response as above.)

Reviewer 3 Report
The authors presented an article titled: “Experimental Investigation on Influence of Process Parameter of Wire Electric Discharge Machining Using Nimonic-263 Super Alloy”. In this paper, the Wire Electric Discharge Machining (WEDM) process variables optimization for the Nimonic-263 superalloy was demonstrated by using the Taguchi-approach, which has multiple performance qualities, including Material Removal Rate (MRR), Surface Roughness (SR), and Kerf Width (KW). In this work, Gap Voltage (GV), Pulse on time (Ton), Pulse off time (Toff), and Wire Feed (WF), are considered as the variable process factors. The article is interesting, however, there are several points in the article that require further explanation.
Comment 1:
Abstract
Abstract can be improved. Present in the abstract novelty, practical significance of presented method. In the abstract, the authors described only what was studied.
Comment 2:
2. Materials and Methods
Describe the WEDM machine in more detail (manufacturer, etc).
Comment 2:
4. Conclusions
Add quantitative and qualitative work results. In addition, it is necessary to more clearly show the novelty of the article and the advantages of the proposed method. What is the difference from previous work in this area? Show practical relevance. Presented conclusions are only a description of the test results. Conclusions should reflect the purpose of the article.
Comment 3:
It will be useful to add a section of Nomenclature in which to sign all the abbreviations encountered in the article. There are many abbreviations in the text and such a section will help to find the description of the necessary element.
For example:
WEDM : the Wire Electric Discharge Machining
MRR : the Material Removal Rate
SR : the Surface Roughness
etc.
Comment 4:
Article is very interesting especially that refers to erosive machining of hard machinable materials used in aerospace industry. All equations are derived correctly and very clearly. The literature is very well chosen. The article can be published in the “Materials” journal after minor corrections.
Minor editing of English language required
Author Response

(The authors gave the same response as above.)

Round 2
Reviewer 1 Report
Accept in present form
Minor editing of English language required
Reviewer 2 Report
The authors did sufficiently address my comments and I therefore reject the manuscript.